# Nanoparticle Coatings on Glass Surfaces to Prevent *Pseudomonas fluorescens* AR 11 Biofilm Formation

**DOI:** 10.3390/microorganisms11030621

**Published:** 2023-02-28

**Authors:** Daniele Marra, Irene Perna, Giulio Pota, Giuseppe Vitiello, Alessandro Pezzella, Giuseppe Toscano, Giuseppina Luciani, Sergio Caserta

**Affiliations:** 1DICMaPI, Università di Napoli Federico II, Piazzale V. Tecchio 80, 80125 Napoli, Italy; 2CSGI, Center for Colloid and Surface Science, Via della Lastruccia 3, 50019 Florence, Italy; 3CEINGE, Advanced Biotechnologies, 80145 Naples, Italy

**Keywords:** biofilm, wetting, hybrid nanoparticles, confocal laser scanning microscopy, cold plasma, antibiofilm, antimicrobial

## Abstract

Microbial colonization of surfaces is a sanitary and industrial issue for many applications, leading to product contamination and human infections. When microorganisms closely interact with a surface, they start to produce an exo-polysaccaridic matrix to adhere to and protect themselves from adverse environmental conditions. This type of structure is called a biofilm. The aim of our work is to investigate novel technologies able to prevent biofilm formation by surface coatings. We coated glass surfaces with melanin-ZnO_2_, melanin-TiO_2_, and TiO_2_ hybrid nanoparticles. The functionalization was performed using cold plasma to activate glass-substrate-coated surfaces, that were characterized by performing water and soybean oil wetting tests. A quantitative characterization of the antibiofilm properties was done using *Pseudomonas fluorescens* AR 11 as a model organism. Biofilm morphologies were observed using confocal laser scanning microscopy and image analysis techniques were used to obtain quantitative morphological parameters. The results highlight the efficacy of the proposed surface coating to prevent biofilm formation. Melanin-TiO_2_ proved to be the most efficient among the particles investigated. Our results can be a valuable support for future implementation of the technique proposed here in an extended range of applications that may include further testing on other strains and other support materials.

## 1. Introduction

The attachment of bacterial cells to a surface is the first step toward the formation of a complex tridimensional structure known as a biofilm. Biofilms are made of bacteria embedded in a matrix of polysaccharides, extracellular DNA, and protein [1,2], allowing for a sessile bacterial growth lifestyle completely different from a free-floating one.

From a rheological point of view, biofilms can be defined as viscoelastic gels in which bacteria cooperate in order to maintain an organized spatial arrangement [3,4]. In fact, to sustain this organization, adhering bacterial cells exhibit different patterns of gene expression [5] and molecular signaling [6], oriented to enhance bacterial resilience towards chemical stresses (e.g., pH changing, environmental temperature, nutrient availability [2]), mechanical stimuli (e.g., shear stress [7,8,9,10,11]), and antimicrobial agent (e.g., antibiotics and detergent solutions [12]). This bacterial persistence on surfaces is usefully employed in the case of wastewater treatment [13] and bioremediation [14], but on the other hand is an issue in many industrial, food processing, and sanitary fields. In industrial plants, biofilms are able to induce biofouling [15], resulting in the clogging phenomena of valve and hydraulic systems with the consequent loss of productivity and enhanced energy demand. In the case of food processing units and medical devices, biofilms can contaminate equipment surfaces causing cross-contamination and risks for human health [16,17]. This phenomena can occur due to the occasional detachment of biofilm filaments, called streamers, due to flow condition [18]. Moreover, several bacterial biofilm are responsible for human infections, such as *Pseudomonas aeruginosa* leading to urinary tract infections through catheter contamination [19].

Due to such negative implications, the prevention of biofilm formation on surfaces is a relevant research topic and is envisaged to bring huge social economic and health benefits [20,21]. Among biofilm eradication strategies, one promising approach is the immobilization of a selected antimicrobial agent [22,23] to create an active coating on surfaces.

Notably, different coatings have been explored so far, including antimicrobial peptides, polycations, polymer brushes [24], and metals (Ag, Zn, Cu) or metal oxide nanoparticles (TiO_2_, ZnO, Fe_2_O_3_, Cu_2_O) [25]. Particularly, the last are bound to exhibit potent biocide performance in view of a high surface to volume ratio [25,26]. Furthermore, the bioinspired design of nanomaterials can be available for a plethora of advanced biocompatible antibiotic systems which mimic natural strategies toward pathogens [27]. In this context, melanins, a class of natural pigments widely available in living organisms, exhibit intrinsic antimicrobial properties [28]. These features can be boosted by combining melanins with an inorganic nanostructure phase, acting as a catalyst and/or morphological agent for melanin formation [29]. Following this synthesis strategy, hybrid melanin-based nanostructures have been obtained with a wide range of antimicrobial activity towards both Gram(+) and Gram(−) strains [30].

Surface functionalization requires stable bonding of the active principle such as nanoparticles to the surfaces of technological relevance, such as glass, plastic, or metal. An innovative and versatile method for the covalent bonding of antimicrobial molecules on different types of surfaces is represented by cold plasma technology. A cold plasma (CP) is a partially or fully ionized gas, depending on the application, that can be used for the deposition of an organic or inorganic film (plasma coating) or for the activation/modification of the surface with or without its functionalization (plasma treatment) [31].

In this work, we propose an antibiofilm coating based on hybrid nanoparticle-melanin agents immobilized using the oxygen CP technique. Surface coating was assessed by the wetting analysis. We selected *Pseudomonas fluorescens* AR11 as the model system of a single-strain biofilm. *P. fluorescens* is a Gram-negative rod-shaped aerobic bacterium known to form biofilms [22]. Biofilm growth was assessed using a confocal laser scanning microscopy (CLSM) after fluorescent staining. Images were processed using standard image analysis techniques. Quantitative parameters were estimated in order to evaluate biofilm biomass, thickness, surface area, and roughness.

The surface functionalization techniques proposed here can be relevant in a wide range of applications, including in the biomedical [32] field and for deep space [20,33,34,35] exploration missions.

## 2. Materials and Methods

### 2.1. Materials

Acetic acid (purity ≥ 99%), isopropanol anhydrous (purity ≥ 99%), ethanol (purity ≥ 99.8%), triethylamine (TEA, purity ≥ 99.5%), titanium isopropoxide (TTiP, purity ≥ 99.9%), and zinc acetate (ZnAc, purity ≥ 99.9%) were purchased from Sigma-Aldrich (Milan, Italy). The synthesis of 5,6-dihydroxyindole-2-carboxylic acid (DHICA) and 5,6-dihydroxyindole (DHI) was performed according to previously described protocols [36]. All reagents were used without further purification.

### 2.2. Synthesis and Characterization of Hybrid Nanoparticles

Hybrid melanin-TiO_2_: Mel/TiO_2_ nanoparticles were prepared through a synthesis methodology based on a hydrothermal route, as previously proposed and described by some authors of this work [36,37,38]. In particular, 3 mL of titanium isopropoxide (TTiP)/isopropanol solution (3.38 M in TTiP) was added dropwise to 15.65 mL of a water/acetic acid solution at pH 1.5. The reaction mixture instantaneously turned from transparent into white due to the formation of a precipitate. The system was kept under stirring for 24 h until all the precipitate was resuspended and a yellowish TiO_2_ colloidal suspension formed. Next, 25 mg of a dry mixture of the melanogenic precursors, DHI and DHICA (3:1 *w*/*w*), were dissolved into the TiO_2_ colloidal suspension to trigger a Ti(IV)-catalyzed eumelanin oxidative polymerization. TiO_2_ nuclei were precipitated by the dropwise addition of triethylamine (TEA) until the pH reached a value of 7. The obtained suspension was then poured into a closed Teflon recipient (up to ¾ of the volume of the reactor), placed into a circulating oven and kept at 120 °C for 24 h. Hybrid nanoparticles were collected by centrifugation and washed 3 times with distilled water. As a reference, bare TiO_2_ nanoparticles were produced following the same protocol without the addition of the melanin monomers.

Hybrid melanin-ZnO: a solvothermal synthesis route previously defined [39] was used to produce bare and hybrid Mel/ZnO nanoparticles as well. Briefly, zinc acetate dihydrate (ZAD, 1.815 g) was dissolved into 101.3 mL of a mixture of TEA and ethanol (2.3% *v*/*v* in TEA). The system was kept under stirring for 15 min to allow for complete dissolution of the zinc precursor. At this time, 32.28 mg of a DHI and DHICA (3:1 *w*/*w*) dry mixture was added to the ZAD/TEA/ethanol solution to induce pigment polymerization. Subsequently, 11 mL of water was poured into the reaction mixture, which lost its transparency and revealed a precipitation phenomena involving the Zn(OH)_2_ nuclei. The system was then transferred into a sealed Teflon reactor, put in a circulating oven and kept at 120 °C for 2 h. Finally, Mel/ZnO nanoparticles were collected by centrifugation and washed three times with distilled water. As a reference, bare ZnO nanoparticles were produced following the same protocol without the step concerning the addition of the melanin monomers.

Crystalline phases of TiO_2_- and ZnO-based nanoparticles were identified by X-ray diffraction (XRD) experiments carried out using a PANalytical diffractometer with a nickel filter and CuKa radiation.

Thermogravimetric analysis (TGA) was carried out by using a simultaneous thermo-analyser SDT Q600 (TA Instrument, New Castle, DE, USA) in order to assess the presence and amount of the organic component within both Mel/TiO_2_ and Mel/ZnO samples. Quantitative analysis was performed according to the standard test methods for proximate analysis [40] and by following a previously described protocol [39].

### 2.3. Glass Surface Functionalization

The functionalization of surfaces was performed through oxygen plasma (Diener Electronic Femto). The reactor is made with a round vacuum chamber in borosilicate glass (95 mm diameter, 280 mm depth, 2 L volume) equipped with an aluminum electrode for the round vacuum chamber. Samples (bare glass slides) were placed inside the chamber for the etching process and the operating parameters are reported here:Pumping down stage: the reactor chamber was closed and the pressure was set at 0.15 mbar through the vacuum pump (Leybold SC5D oil-free);Gas supply stage: the reactor chamber was fed with pure oxygen gas for 1 min and the pressure was set at 0.5 mbar;Plasma activation: oxygen plasma was ignited by applying 20 W and the plasma was kept active for 2 min;Flushing stage: the chamber was flushed with oxygen gas for 10 s;Venting stage: the chamber internal pressure was restored at room pressure.

After the surfaces’ activation, the coating was carried out by a drop-casting procedure. A 600 μL droplet of an aqueous suspension of nanoparticles at a concentration of 1.5 mg/mL was deposited in order to homogeneously cover the entire surface of the glass slide that was previously treated. Subsequently, the coated glass slides were dried in a ventilated oven at a temperature of 60 °C for about 1 h. This process was performed for each aqueous suspension of melanin-TiO_2_, melanin-ZnO, and TiO_2_.

### 2.4. Sample Wetting Analysis

The wetting properties of the coated and uncoated glass samples were investigated in order to analyze both the hydrophilic and hydrophobic behaviors of the surfaces. Specifically, soybean oil and distilled water drops were used to measure the static contact angles. Droplets of 30 μL were placed on the coated surface of each glass slide and allowed to equilibrate for 30 s. Side view images of each droplet was obtained through a Canon EOS 60 D-S equipped with Canon Macro Lens EFS 60 mm. The images were analyzed using a commercial image analysis software (Image ProPlus 6.0) in order to measure the left and right contact angle values of the sessile droplets. The wetting measurement was repeated for 3 different samples for each treatment. More than one water and oil droplet (typically 2 for each liquid) was placed in a different position on each sample investigated. The data reported are calculated as the mean over the minimum 12 contact angle measurements for each condition and the error bars are calculated as the standard error of the mean.

### 2.5. Microorganism and Culture Conditions

Suspended cultures of *Pseudomonas fluorescens* strain AR 11 [41,42,43] (DSMZ, Braunschweig, Germany) were grown in a M9-minimal medium (6.8 g/L of Na_2_HPO_4_, 3 g/L of KH_2_PO_4_, 0.5 g/L of NaCl, and 1 g/L NH_4_Cl, 0.24 g/L MgSO_4_·7H_2_O, 0.04 g/L CaCl_2_·2H_2_O, 0.05 g/L EDTA, 8.3 mg/L FeCl_3_, 0.84 mg/L ZnCl_2_, 0.1 mg/L CuCl_2_·2 H_2_O, 0.1 mg/L CoCl_2_·2H_2_O, 0.1 mg/L H_3_BO_3_, and 0.016 mg/L MnCl_2_·4 H_2_O supplemented with 0.4% succinic acid at pH 7) at 30 °C in a shaken vessel (90 rpm). Growth was followed until an optical density OD_600nm_ ≈ 0.5 was achieved.

### 2.6. Biofilm Growth Conditions

Biofilms were grown on microscope glass slides 76.2 × 25.4 × 1 mm^3^ (raw glass and nanoparticles functionalized) placed inside 90 mm polystyrene Petri dishes (PROMED, Italy) dipped in a stagnant bacterial suspension according to the protocol of Castigliano et al. [44]. Glass coupons were first cleaned in a 95% ethanol solution for 30 min, rinsed with distilled water, and finally oven-dried at 200 °C for 2 h. After this sterilization procedure, a microscope slide was placed horizontally in each Petri dish and submerged with 15 mL of a bacterial suspension obtained by 1:10 dilution of a fresh culture with sterile minimal medium. This experimental procedure was adopted to ensure aerobic conditions for bacterial growth. Biofilm formation and development was carried out at 30 °C for 48 h.

### 2.7. Biofilm Imaging and Morphology Analysis

Biofilms were stained for subsequent acquisition by confocal scanning laser microscopy (CSLM) using the commercial kit, LIVE/DEAD™ (Biofilm Viability Kit, Molecular Probes, Invitrogen, Carlsbad, CA, USA). The kit uses mixtures of the SYTO^®^ 9 green fluorescent nucleic acid stain for live bacterial cell and the red fluorescent nucleic acid stain, propidium iodide, for dead ones. The excitation/emission maxima for these dyes are approximately 480/500 nm for the SYTO^®^ 9 stain and 490/635 nm for propidium iodide. Staining solutions were prepared by adding 3 μL of SYTO^®^ 9 and 3 μL of propidium iodide to 1 mL of distilled water. Biofilm-coated glass slices were stained by placing 200 μL of staining solution on the sample surface in the dark for 30 min at room temperature. In the protocol, subsequent washing was avoided to prevent uncontrolled biofilm detachment [44]. Images were acquired by using a confocal laser scanning microscope (LSM 5 Pascal, Zeiss) equipped with a helium/neon laser (LASOS Lasertechnik GmbH, LGK SAN7460A). Experimental observations were performed with a Plan Apo λ 63 X/1.49 NA oil-objective and a Nikon digital camera. Excitation was provided at a wavelength of 488.6 nm using a detection filter of 498 nm and at a wavelength of 561.5 nm with a detection filter of 580 nm. The Z-stacks of the samples were acquired along the vertical direction using a 0.6 μm interval. Image processing was performed by COMSTAT2 [7,45], a MATLAB© script, developed to be run by Image J. Comstat2 is a program that is able to analyze the image stacks of biofilms, recorded by confocal microscopes. It can measure several quantitative parameters by dividing the Z-stack acquisition images in a volume element called voxels. The width (x) and length (y) of the voxel are the side lengths of the pixel and the height (z) is the distance between the slices.

Both green (live) and red (dead) channels were processed. Several parameters were estimated:Biomass (μm^3^/μm^2^): defined as the volume of biomass per unit area and estimated as the volume of all voxels that contain biomass divided by the substratum area; COMSTAT2 counts as biomass all voxels above a given threshold;Average thickness (biomass) (µm): considering only the area covered by the biomass;Average thickness (entire area) (µm): considering the entire area of the stack;Maximum thickness (μm): evaluated as the highest point of the biofilm over the substratum;Roughness coefficient, Ra (nondimensional): measuring the variability in height of the biofilm [46];Surface area (μm^2^): calculated as the area summation of all biomass voxel surfaces exposed to the background and area occupied in the layer (μm^2^): considering biomass pixels in each layer (confocal slice).

To verify the statistical significance of the results in our work, we report (Table 1) measurements as the mean ± standard deviation calculated from 3 independent experiments. For each experiment, 9 image stacks were taken.

## 3. Results and Discussion

### 3.1. Properties of Hybrid Nanoparticles

The formation of Mel/TiO_2_ and Mel/ZnO nanoparticles was verified by XRD and TGA measurements. As shown by the XRD spectra in Figure 1, both Mel/TiO_2_ and Mel/ZnO nanoparticles presented a crystalline structure, typical of anatase and wurtzite phases, respectively. The spectra agree with those previously obtained for both kinds of hybrid nanoparticles [37,38], thus confirming the formation and growth of the inorganic fraction of nanoparticles.

TGA curves (Figure 2A,B) showed two major weight losses in the ranges of 700–890 °C and 90–670 °C for Mel/TiO_2_ and in the ranges of 490–760 °C and 360–450 °C for Mel/ZnO, which are related to the decomposition of the organic fraction. The quantitative analysis estimated the melanin content to be about 8% *w*/*w* for Mel/TiO_2_ and 16% *w*/*w* for Mel/ZnO, respectively.

### 3.2. Surface Wetting Analysis

The wetting analysis of the surfaces is reported in Figure 3. On the left, side views of 30 μL droplets deposited on bare and functionalized samples are reported for qualitative comparison. On the right, the values of the contact angles measured by image analysis are reported for the different samples investigated.

The contact angles measured on bare glass are in agreement with the measures reported in previous work [7]. The plasma-treated samples without nanoparticle functionalization were completely wetted by water (no measurable CA) whereas the contact angles of the soybean oil droplets were similar to those on bare glass. TiO_2_ showed a significant reduction of both water and soybean oil CA that had a comparable value of about 10° proving a high hydrophilic and oleophilic behaviour. Also in the case of melanin/ZnO, water and oil showed a similar behavior with the contact angle being slightly higher (about 20°), comparable with the case of oil on bare glass. The same value of about 20° was measured for water on the melanin/TiO_2_ samples while in this case, the surface was completely wetted by soybean oil droplets (no measurable CA).

### 3.3. Biofilm Morphology on Different Functionalized Samples

After 48 h of growth on bare glass and different functionalized samples (melanin/ZnO, melanin/TiO_2_, and TiO_2_), *Pseudomonas fluorescens* biofilms were stained with the LIVE/DEAD kit before CLSM acquisitions. For each type of coating, three different samples were analyzed and for each sample, a 3D mosaic scan [47] (3 × 3 × 15–20 layers, depending on biofilm thickness) was acquired. Using biofilm growth on bare glass as the control experiment, the effect of the nanoparticle coatings is shown by comparing the functionalized samples with the unfunctionalized one (a single field of view of mosaic scan is reported in Figure 4).

Figure 5 can provide a better understanding of the 3D structure. For each sample, all the Z planes of the image stack are collapsed into a single plane from the orthogonal and side viewpoints. It is possible to qualitatively visualize the effects of the different types of functionalization on biofilm formation. The surfaces coated with Mel/ZnO appear to be similar to the biofilm growth on bare glass while the TiO_2_ and Mel/TiO_2_ coatings appear to negatively affect bacterial colonization at the surfaces.

Bare glass and melanin/ZnO clearly show similar thickness and surface coverage while TiO_2_ coating appears to affect live cell density. Concerning the experiments using Mel/TiO_2_ coatings, the CSLM side and orthogonal projections show a spatial pattern of biofilm growth differing from that on bare glass. In this case, the cells aggregate into heterogeneously distributed clumps along the surface; moreover, the biofilm appears to be less thick.

To quantitatively characterize these effects, confocal stack images were analyzed using COMSTAT2 to obtain specific quantitative parameters of biofilm morphology: cell biomass, average biofilm thickness (considering only biofilm area and all the surface area), surface area, and roughness coefficient. Table 1 reports the values of the different morphological parameters averaged over at least three independent experiments.

According to the data analysis, melanin/ZnO coating compared with bare glass has no antibiofilm activity, showing instead a slight increase in biofilm biomass and thickness. This increase has the same magnitude for both live and dead cells. The other two coatings tested, TiO_2_ and melanin/TiO_2_, exhibit a strong contact killing effect [20] as can be deduced from the live biomass values. As evidence of this effect, while for bare glass and melanin/ZnO live cells represent on average 30–40% of all the biomass, in the oxide titanium coatings the percentage is less than 1% (approximately 0.05% for TiO_2_ and 0.1% for melanin/TiO_2_). While both TiO_2_ and melanin/TiO_2_ coated surfaces exhibit a similar contact killing effect, the hybrid particle coating show an improved antiadhesion property, as shown by comparison of the dead biomass and average thickness. In the melanin/TiO_2_ coating, both live and dead biomasses are reduced while the TiO_2_ coating affects only live cell biomass. As evidence of this effect, the average biofilm thickness considering the entire area of the sample is the lowest in the melanin/TiO_2_ coating.

Another useful parameter for the analysis of the 3D structure of biofilm growth on different coatings is the quantification of the area occupied by live and dead bacteria in each layer of the CLSM Z-stack acquisition, as is reported in Figure 6.

A comparison between the area occupied by live and dead cells shows that dead cells occupy a greater area in all the cases of investigation, which is in agreement with the biomass values in Table 1. Moreover, the melanin/ZnO area for both live and dead cells shows similar trends to the ones on bare glass (as further evidence of its inefficiency as an antibiofilm agent for this bacteria). The live cell coverage area values for TiO_2_ and melanin/TiO_2_ coatings are drastically reduced (confirming the contact-killing hypothesis). The dead cell coverage area for the melanin/TiO_2_ coating is also drastically reduced, confirming the anti-adhesive properties.

## 4. Conclusions

We propose a novel methodology to create antibiofilm coatings using cold plasma treatment and hybrid nanoparticles. Cold plasma treatment has the advantage of avoiding the need for potentially noxious chemical linkers between the substratum and the antibiofilm agent, and can be used for a wide range of substrata. In addition, we propose an innovative type of hybrid nanoparticle combining both inorganic and organic antimicrobial compounds as the antibiofilm agent. In this work, three different nanoparticle coatings (Mel/ZnO, Mel/TiO_2_, and TiO_2_) for biofilm prevention were investigated. The antibiofilm properties of these nanoparticle coatings were assessed by quantifying biofilm morphology using CLSM image analysis techniques.

In particular in this work, *Pseudomonas fluorescens* AR 11 biofilms on bare glass and different functionalized samples (Mel/ZnO, Mel/TiO_2_, and TiO_2_) showed different 3D structures depending on the coating types. Melanin/ZnO showed virtually no antibiofilm properties while TiO_2_ coating exhibited apparently only a bacterial killing action without preventing biofilm formation on surfaces. Melanin/TiO_2_ functionalization exhibited the highest antibiofilm properties with an antiadhesion and bacterial killing action. Wetting analysis proved all the samples had comparable omniphilic behaviour with the peculiarity of almost perfect oil wetting (no measurable contact angle) in the case of melanin/TiO_2_-coated surfaces. This result may suggest a connection with the strong antiadhesion properties, but will require further more detailed investigations.

This work provides proof the methodology reported here can represent a relevant support to the design of industrial antibiofilm surfaces. The coating technique presented here can be potentially applied to a wide range of materials (e.g., relevant industrial surfaces) and this methodology can be applied to investigate hybrid nanoparticle antibiofilm properties on other strains as well.

## Figures and Tables

**Figure 1 microorganisms-11-00621-f001:**
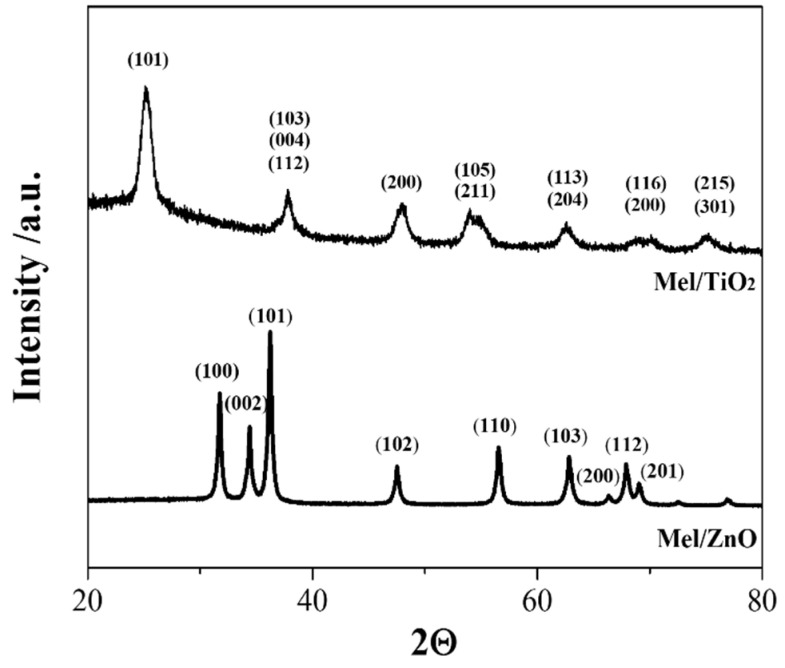
XRD spectra of Mel/TiO_2_ and Mel/ZnO nanoparticles.

**Figure 2 microorganisms-11-00621-f002:**
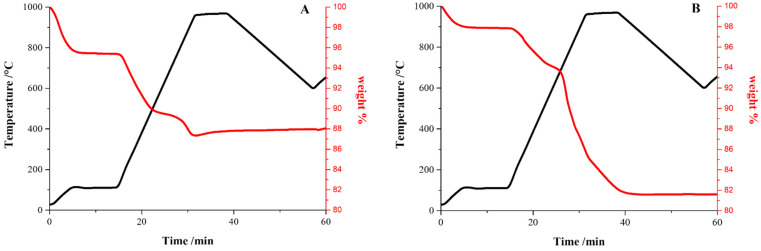
TGA curves of Mel/TiO_2_ (**A**) and Mel/ZnO (**B**) nanoparticles.

**Figure 3 microorganisms-11-00621-f003:**
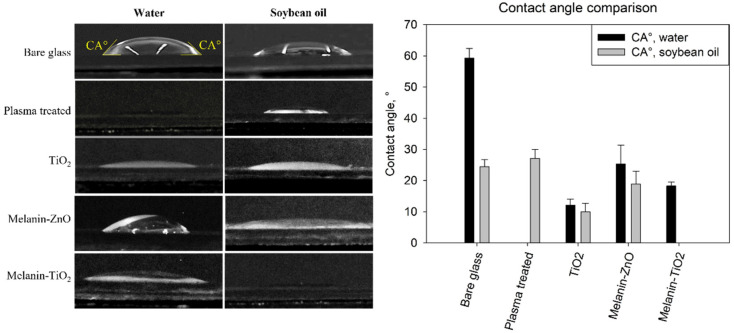
Side views of 30 μL droplets of water and soybean oil. Experiments are conducted in triplicates for each type of sample (nanoparticles of melanin/ZnO, TiO_2_, and melanin/TiO_2_).

**Figure 4 microorganisms-11-00621-f004:**
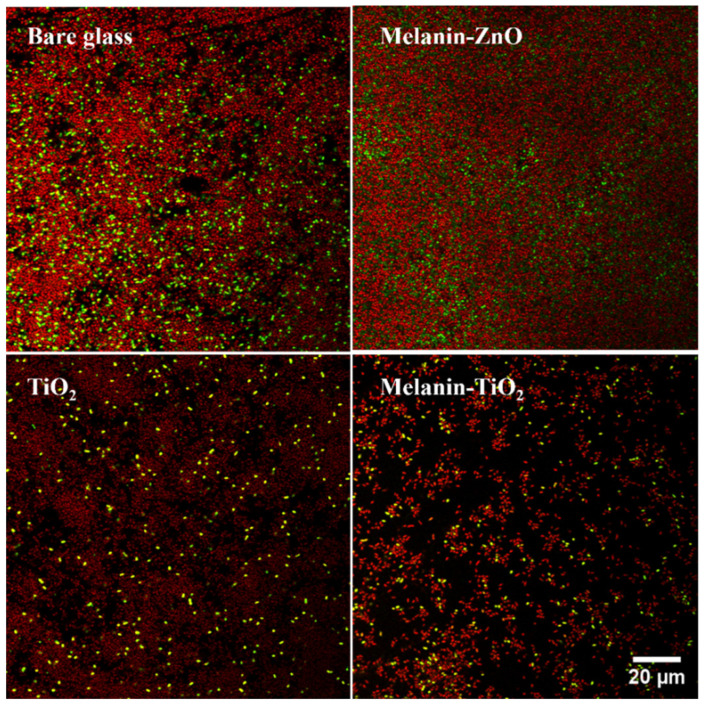
Confocal laser scanning microscopy scans performed with LIVE/DEAD staining of 48 h biofilm growth on bare glass (control) and functionalized samples (nanoparticles of melanin/ZnO, TiO_2_, melanin/TiO_2_). Live bacteria are reported in green, dead in red.

**Figure 5 microorganisms-11-00621-f005:**
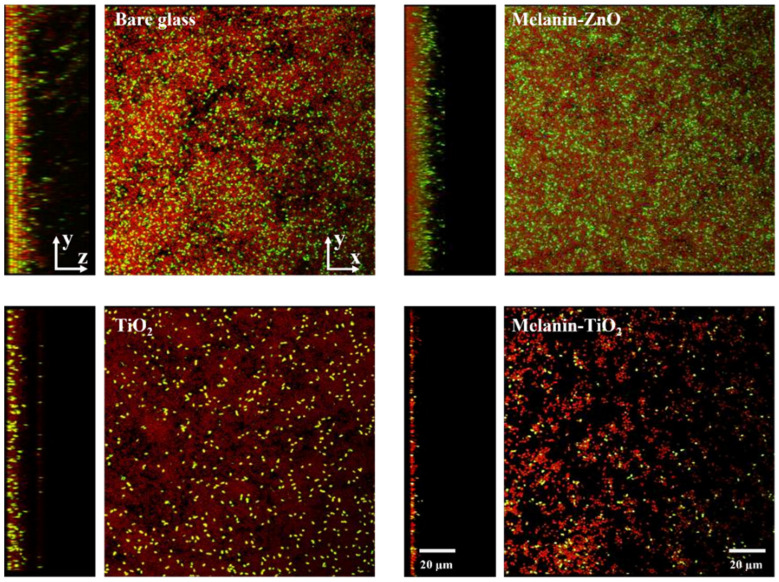
Confocal laser scanning microscopy Z-scan projections performed with live/dead staining of 48h biofilm growth on bare glass (control) and functionalized samples (nanoparticles of melanin-ZnO, TiO_2_, melanin TiO_2_). For each sample, top view (**right**) and side view (**left**) display overlaid images. Live bacteria are reported in green, dead in red.

**Figure 6 microorganisms-11-00621-f006:**
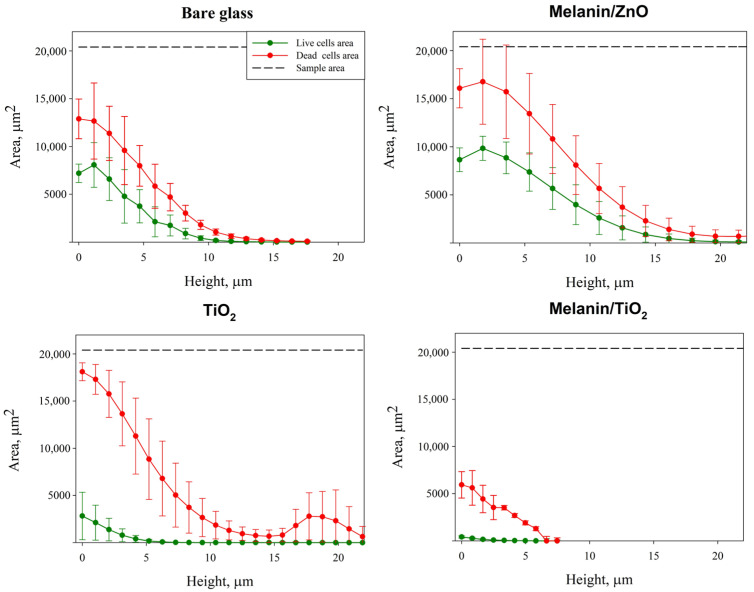
Area occupied in each layer of the Z-scan acquisition by dead cells (in red) and live cells (in green). The grey line represents the total surface area. The height represents biofilm thickness starting from the bottom surface layer to the last biofilm layer detected for each sample. Area is evaluated by processing CSLM stack images of biofilms cultivated under different coatings: bare glass, melanin/ZnO, TiO_2_, melanin/TiO_2_. Average values ± standard deviation calculated from at least triplicate independent experiments are shown.

**Table 1 microorganisms-11-00621-t001:** Biomass, maximum thickness, mean thickness, roughness, and surface area of biofilm growth on different samples. Analysis was performed using COMSTAT. Values are mean ± standard deviation of data from 3 separate experiments. For each experiment, 9 image stacks were taken.

*P. fluorescens* AR11	Bare Glass		Melanin/ZnO		TiO_2_		Melanin/TiO_2_	
	Live	Dead	Live	Dead	Live	Dead	Live	Dead
Biomass (µm^3^/µm^2^)	2.91 ± 0.78	5.99 ± 1.30	6.42 ± 2.81	10.37 ± 2.71	0.48 ± 0.68	8.81 ± 0.80	0.14 ± 0.01	1.63 ± 0.37
Mean thickness (Biomass) (µm)	6.42 ± 1.20	8.60 ± 0.94	10.67 ± 3.85	12.98 ± 3.17	4.40 ± 2.17	11.15 ± 2.01	3.36 ± 0.60	5.50 ± 1.25
Mean thickness (Area) (µm)	4.94 ± 1.31	7.70 ± 1.37	9.55 ± 4.68	12.40 ± 3.35	2.94 ± 4.15	10.71 ± 2.45	0.24 ± 0.03	3.36 ± 1.05
Maximum thickness (µm)	20.70 ± 4.18	23.75 ± 6.24	24.26 ± 3.22	25.20 ± 3.56	5.86 ± 4.64	24.08 ± 1.14	6.54 ± 0.52	8.71 ± 1.88
Roughness (–)	0.74 ± 0.28	0.49 ± 0.18	0.65 ± 0.36	0.42 ± 0.23	0.67 ± 0.37	0.37 ± 0.01	1.88 ± 0.03	0.85 ± 0.22
Surface Area (10^5^ µm^2^)	2.90 ± 0.71	3.66 ± 0.33	4.86 ± 1.27	5.08 ± 1.72	0.62 ± 0.87	4.67 ± 0.31	0.21 ± 0.02	1.81 ± 0.44

## Data Availability

Raw data will be made available on request.

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
