# Peer review of "Nanoparticle Coatings on Glass Surfaces to Prevent Pseudomonas fluorescens AR 11 Biofilm Formation"

_microorganisms, 2023, doi:10.3390/microorganisms11030621_

Round 1

Reviewer 1 Report

The manuscript entitled: "Nanoparticle coatings on glass surfaces to prevent biofilm formation" aimed to investigate the ability of nanoparticle coating to prevent biofilm formation on surfaces.

Please consider the following comments:

1- The abbreviations should be mentioned in complete words when it is first mentioned in the manuscript. 

2- The abstract should be more comprehensive.

3- Less number of keywords should be mentioned

4- Recent references related to the manuscript should be cited like:

Attallah, N. G., Elekhnawy, E., Negm, W. A., Hussein, I. A., Mokhtar, F. A., & Al-Fakhrany, O. M. (2022). In vivo and in vitro antimicrobial activity of biogenic silver nanoparticles against Staphylococcus aureus clinical isolates. Pharmaceuticals15(2), 194.

5- Why wasn't the effect of the nanoparticles studied genotypically (using qRT-PCR for example)?

6- What about the utilized statistical analysis? it isn't mentioned in the manuscript.

7- There is no details concerning COMSTAT 

Reviewer 2 Report

This study aims at the to investigation of technologies for prevention of biofilm formation by surface coatings. The Authors proposed an antibiofilm coating based on hybrid nanoparticle-melanin agents immobilized using the oxygen cold plasma technique.

The paper is written within the Journal required style. The whole study was seriously conducted and methodology applied is scientifically sound (with the exceptions described below). The manuscript is well-designed, results and discussion are plausible and coherent according to the scientific standards; they are also explicatory and sufficiently presented with several well-prepared figures and tables. The objective of the study is well defined and the Authors explain how they reached the conclusions (although I have some comments on the conclusions drawn). The paper is also written quite correctly in terms of its language.

The only comment I have, concerns the number of bacteria investigated. The Authors used just one bacterial species (simultaneously just one strain of this species). Therefore, the title of the manuscript and the conclusions drawn should be limited only to that species (e.g., Nanoparticle coatings on glass surfaces to prevent Pseudomonas fluorescens AR 11 biofilm formation). Nevertheless, the use of only one species and strain of bacteria makes the properties of the developed material unreliable and relatively easy to undermine. For this reason, I recommend performing microbiological analyzes using other species of microorganisms (at least one gram-positive bacteria and Candida spp.). Biofilm assessment should also be performed using at least one more test. Analysis using propidium iodide and Syto-9 staining involves washing steps that may cause uncontrolled detachment of the biofilm. For example, one test should be performed showing the amount of biofilm over the entire surface of the material. SEM imaging would also provide valuable results.

Round 2

Reviewer 2 Report

The manuscript can be published in its present form.